# MEF2C regulates cortical inhibitory and excitatory synapses and behaviors relevant to neurodevelopmental disorders

**Adam J Harrington[1,2†], Aram Raissi[2†], Kacey Rajkovich[3], Stefano Berto[3], Jaswinder Kumar[2,4], Gemma Molinaro[3], Jonathan Raduazzo[2], Yuhong Guo[2], Kris Loerwald[3], Genevieve Konopka[3], Kimberly M Huber[3], Christopher W Cowan[1,2*]**

[1]Department of Neurosciences, Medical University of South Carolina, Charleston, United States; [2]Department of Psychiatry, Harvard Medical School, Belmont, United States; [3]Department of Neuroscience, The University of Texas Southwestern Medical Center, Dallas, United States; [4]Medical Scientist Training Program, The University of Texas Southwestern Medical Center, Dallas, United States

**Abstract** Numerous genetic variants associated with *MEF2C* are linked to autism, intellectual disability (ID) and schizophrenia (SCZ) – a heterogeneous collection of neurodevelopmental disorders with unclear pathophysiology. MEF2C is highly expressed in developing cortical excitatory neurons, but its role in their development remains unclear. We show here that conditional embryonic deletion of *Mef2c* in cortical and hippocampal excitatory neurons (Emx1-lineage) produces a dramatic reduction in cortical network activity in vivo, due in part to a dramatic increase in inhibitory and a decrease in excitatory synaptic transmission. In addition, we find that MEF2C regulates E/I synapse density predominantly as a cell-autonomous, transcriptional repressor. Analysis of differential gene expression in *Mef2c* mutant cortex identified a significant overlap with numerous synapse- and autism-linked genes, and the *Mef2c* mutant mice displayed numerous behaviors reminiscent of autism, ID and SCZ, suggesting that perturbing MEF2C function in neocortex can produce autistic- and ID-like behaviors in mice.

*For correspondence: cowanc@musc.edu

†These authors contributed equally to this work

Competing interests: The authors declare that no competing interests exist.

## Introduction

An imbalance of excitatory and inhibitory synaptic transmission in the brain is an emerging theory of the pathophysiology of multiple neurodevelopmental and neuropsychiatric disorders (*Garber, 2007*; *Zoghbi, 2003*), including autism and SCZ. However, the genes and molecules that regulate the number of excitatory and inhibitory synapses formed and maintained on neurons remain poorly understood.

The MEF2 transcription factor genes are expressed in both excitatory and inhibitory neurons throughout development and adulthood in overlapping, but unique, expression patterns (*Lyons et al., 2012*; *Shalizi and Bonni, 2005*; *McKinsey et al., 2002*), and they have been shown to regulate excitatory synapse density on multiple neuron types (*Flavell et al., 2006*; *Li et al., 2008*; *Barbosa et al., 2008*; *Pulipparacharuvil et al., 2008*). For example, MEF2A and MEF2D can regulate activity-dependent elimination of glutamatergic synapses on both hippocampal pyramidal neurons and medium spiny neurons of the striatum in a cell-autonomous manner (*Flavell et al., 2006*; *Pulipparacharuvil et al., 2008*). Expression of a constitutively-active form of MEF2C (MEF2C-VP16) promotes excitatory synapse elimination in hippocampal pyramidal neurons in a complex process

**eLife digest** Abnormal development of the brain contributes to intellectual disability, as well as to a number of psychiatric disorders, including schizophrenia and autism. As the brain develops, neurons establish connections with one another called synapses, which are either excitatory or inhibitory. At excitatory synapses, an electrical signal in the first cell increases the likelihood that the second cell will also produce an electrical signal. At inhibitory synapses, electrical activity in the first cell reduces the chances of the second cell producing an electrical signal. An imbalance between excitatory and inhibitory activity is one of the factors thought to give rise to neurodevelopmental disorders.

Many individuals with schizophrenia, autism or intellectual disability possess mutations in, or near, a gene called *MEF2C*. This gene, which is active in both excitatory and inhibitory neurons, encodes a protein that regulates the activity of many other genes during brain development. Harrington, Raissi et al. therefore hypothesized that alterations in *MEF2C* might predispose individuals to neurodevelopmental disorders by disrupting the balance of excitatory and inhibitory synapses in the developing brain.

To test this idea, Harrington, Raissi et al. generated mice that lack the *Mef2c* gene in a large proportion of their neurons throughout development. As predicted, the animals showed an imbalance of excitatory and inhibitory synapses in the brain's outer layer, the cortex. They also displayed changes in behavior like those seen in autism. These included a loss of interest in social interaction and a reduction in vocalizations, suggesting impaired communication. Other behavioral changes included hyperactivity, repetitive movements and severe learning impairments: all features commonly observed in human neurodevelopmental disorders.

The next challenge is to understand when, where and how MEF2C acts in the cortex to shape the balance of excitatory and inhibitory connections. Once this is known, further studies can test whether disrupting these processes leads directly to behaviors characteristic of autism, schizophrenia and intellectual disability. This may lead to the development of new drugs that can reverse or improve the symptoms of these conditions in affected individuals.

that requires the RNA-binding protein, Fragile X mental retardation protein (FMRP) (*Flavell et al., 2006*; *Pfeiffer et al., 2010*; *Tsai et al., 2012*; *Wilkerson et al., 2014*).

Brain-wide deletion of *Mef2c* was reported to cause an increase in dendritic spine density on dentate granule neurons of the hippocampal dentate gyrus (*Barbosa et al., 2008*), whereas another group reported that *Mef2c* deletion in embryonic neural stem cells (nestin-Cre), caused deficits in cortical neuron migration and excitatory synaptic transmission in a subset of animals (*Li et al., 2008*). Recent genetic studies have linked human *MEF2C* to a syndromic form of intellectual disability with autistic features, and single-nucleotide polymorphisms (SNPs) near *MEF2C* produce significant risk for SCZ (*Paciorkowski et al., 2013*; *Mikhail et al., 2011*; *Novara et al., 2010*; *Le Meur et al., 2010*; *Cardoso et al., 2009*; *Engels et al., 2009*), which highlight the importance of this gene for normal brain development and function. However, the functional role(s) of MEF2C in early neuronal development, particularly in the neocortex, remains unclear. In the central nervous system, MEF2C is highly expressed very early in brain development (~E11.5), and its expression is enriched in differentiated forebrain neurons within the neocortex and dentate gyrus (*Lyons et al., 1995*; *Leifer et al., 1993*, *1997*). Here, we sought to evaluate the role of MEF2C in differentiated cortical excitatory neurons, and to determine whether loss of MEF2C function in these neuronal populations might produce behavioral and synaptic phenotypes with potential relevance to its associated neurodevelopmental disorders.

## Results

### Generation of *Mef2c* conditional knockout (*Mef2c* cKO) mice

*Mef2c* mRNA is enriched in the developing cortical plate, mature cortex and dentate gyrus (*Leifer et al., 1993*). Immunostaining of mature brain slices with MEF2C-specific monoclonal

antibodies revealed that >99% of the MEF2C-positive cells co-localized with the neuronal marker, NeuN (*Figure 1A*), indicating that MEF2C expression in the cortex is primarily restricted to neurons. To generate conditional gene disruption of *Mef2c* selectively in differentiated forebrain excitatory neurons, we bred homozygous floxed *Mef2c* mutant mice (*Lin et al., 1997*) with mice heterozygous for Cre recombinase inserted into the endogenous *Emx1* gene (*Iwasato et al., 2008*), which expresses Cre in ~90% of differentiated neocortical and hippocampal excitatory neurons and in some forebrain glia starting as early as embryonic day 11.5 (E11.5). The *Mef2c* cKO (*Mef2c^{fl/fl}*; *Emx1^{IRES-Cre}/^+*) mice show selective and dramatic reduction of MEF2C protein levels throughout the cortex and hippocampus, but no reductions were observed in the striatum or thalamus (*Figure 1B*) – Emx1-negative regions that express low levels of MEF2C.

The *Mef2c* cKO offspring were viable and healthy, and their body weights, growth trajectories and Mendelian frequency appear indistinguishable from their Cre-negative littermates (*Figure 1—figure supplement 1A* and data not shown). Using a 3XMRE (MEF2 response element)-luciferase reporter in cultured cortical neurons from control or *Mef2c* cKO offspring, we detected a ~35% decrease in basal MEF2 activity, but no deficit in depolarization-induced, MEF2-dependent transcriptional activity (*Figure 1—figure supplement 1B*), suggesting that MEF2A and MEF2D are sufficient to mediate normal levels of depolarization-dependent MEF2 activity. In young adult *Mef2c* cKO mice, we observed normal gross brain morphology and cortical layer organization (*Figure 1—figure supplement 1C,D*), but *Mef2c* cKO mice did exhibit a slight decrease (~10%) in neocortical thickness compared to controls (*Figure 1C*).

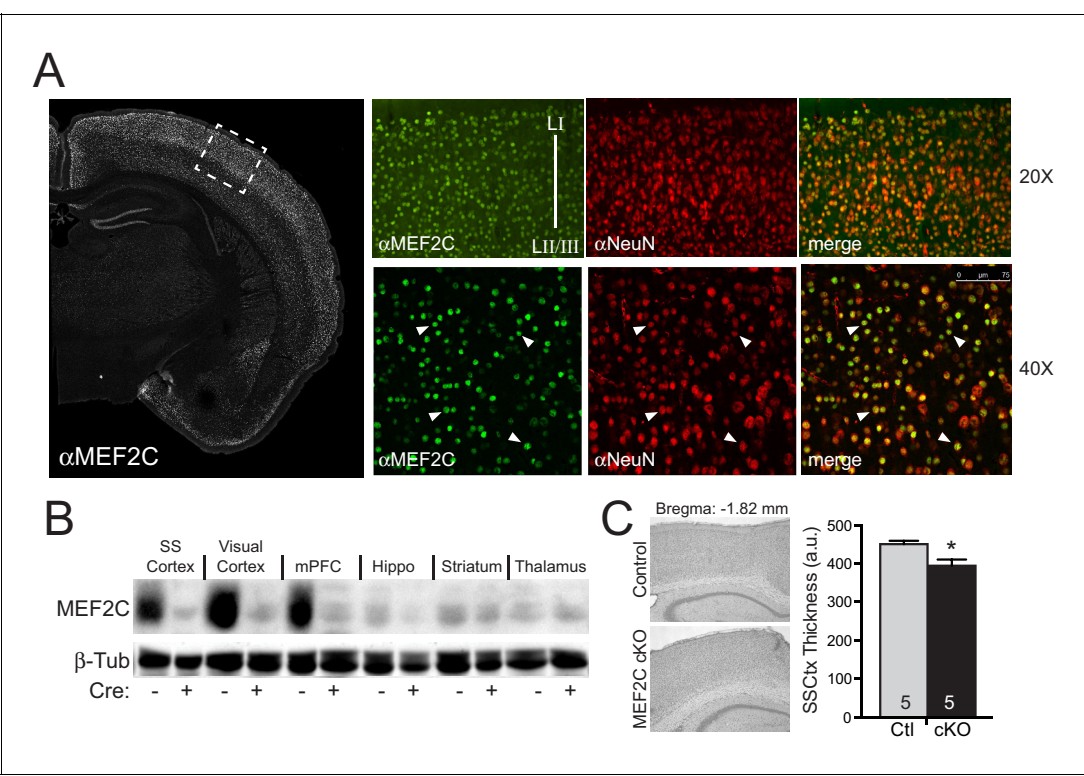

**Figure 1.** Generation of *Mef2c* cKO mice. (A) MEF2C protein (green) is enriched in NeuN-positive cortical neurons (red). (B) Western blot of MEF2C in various brain regions. (C) Somatosensory cortical thickness was slightly reduced in *Mef2c* cKO brains (~10%) compared to control littermates. Thickness was averaged over 4 slices/brain from 5 brains per genotype. Data are represented as mean ± SEM. Statistical significance was determined by unpaired t-test. *p<0.05, ns=not significant.

The following figure supplement is available for figure 1:

**Figure supplement 1.** Neuronal characterization of *Mef2c* cKO mice.

## Increased inhibitory and decreased excitatory synaptic transmission on *Mef2c* cKO cortical neurons

MEF2 transcriptional activity promotes excitatory synapse and dendritic spine elimination in the hippocampus (*Flavell et al., 2006*; *Pfeiffer et al., 2010*; *Tsai et al., 2012*; *Zang et al., 2013*). Therefore, we sought to test whether loss of *Mef2c* in the cortex alters cortical synaptic transmission in vivo. To achieve this, we measured cortical UP states, which are spontaneous, synchronous oscillations of neocortical networks that are driven by recurrent excitatory and inhibitory synaptic circuitry (*Hays et al., 2011*; *Gibson et al., 2008*), to assess overall synaptic function and excitability of the neocortical circuit within the somatosensory cortex (SSC) of the *Mef2c* cKO mice. Surprisingly, we observed large reductions (~90%) in the frequency of spontaneous UP states in acute slices from the SSC of *Mef2c* cKO mice (*Figure 2A*). In addition, the UP states in the *Mef2c* cKO mice were shorter in duration (~50%) and smaller in amplitude (~50%) (*Figure 2A*). To further explore this decrease in neocortical circuit activity, we performed patch-clamp recordings of layer 2/3 pyramidal neurons

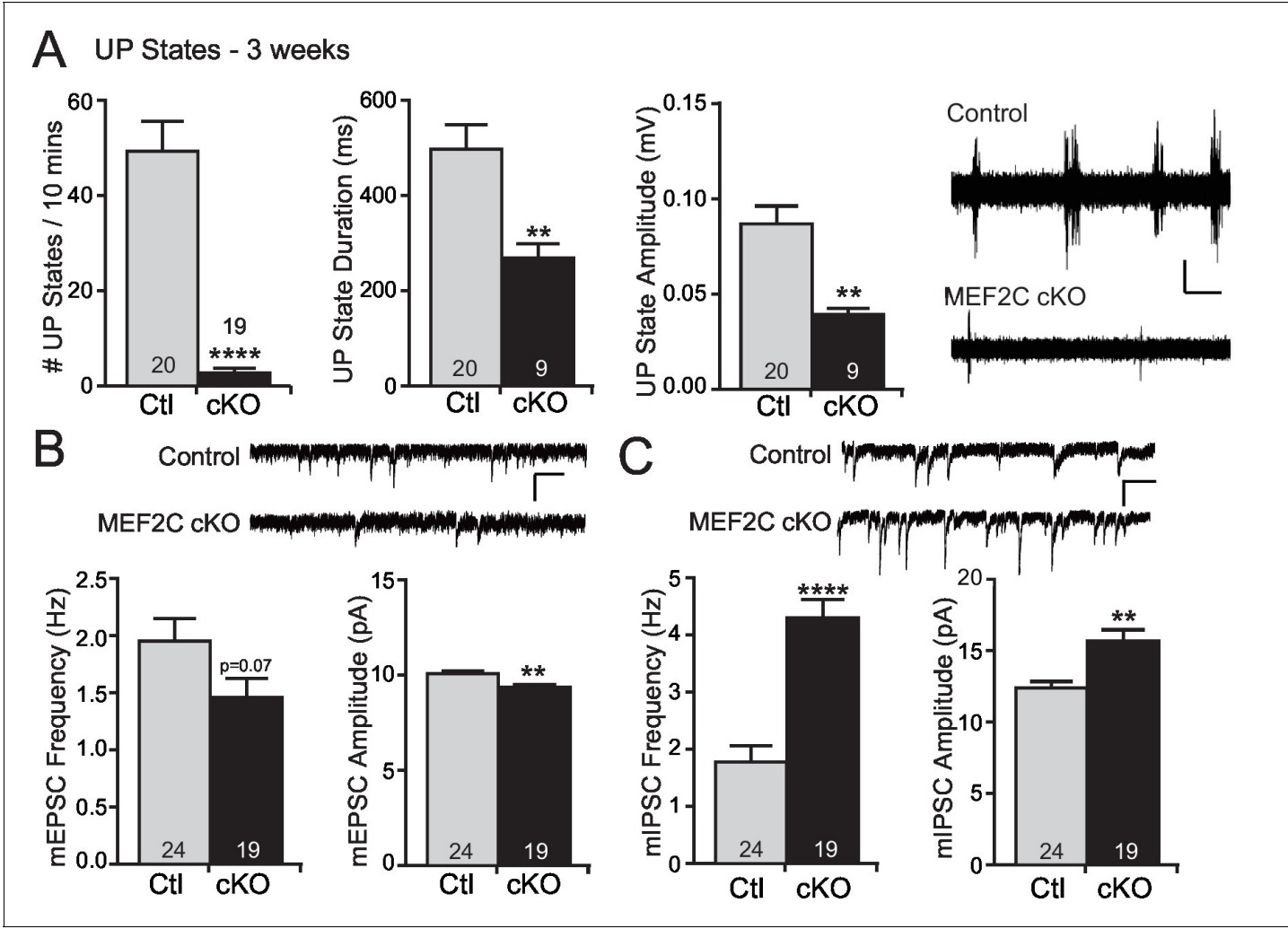

**Figure 2.** Increased cortical inhibition in *Mef2c* cKO mice. (A) UP states in 3-week old *Mef2c* cKO mice. *Mef2c* cKO mice have fewer spontaneous UP states than control mice. Additionally, the duration and amplitude of each spontaneous UP state was significantly reduced in the *Mef2c* cKO mice. Representative recordings from control and *Mef2c* cKO organotypic slices. Scale bar = 50 μV/1 s. (B) *Mef2c* cKO mice have reduced mEPSC frequency and amplitude in cortical layer 2/3 pyramidal neurons from 3-week old mice. Scale bar = 200 ms/10 pA. (C) *Mef2c* cKO mice have increased mIPSC frequency and amplitude in cortical layer 2/3 pyramidal neurons from 3-week old mice. Scale bar = 200 ms/10 pA. Data are represented as mean ± SEM. Statistical significance was determined by unpaired t-test using GraphPad Prism. *p<0.05, **p<0.01, ****p<0.0001. Numbers of slices/neurons (n) are reported in each bar for respective experiment.

from the SSC acute slices. In the *Mef2c* cKO slices, we detected small decreases in both the frequency and amplitude of miniature excitatory postsynaptic currents (mEPSCs) (*Figure 2B*), although the decrease in frequency did not quite reach statistical significance (p=0.07). We also observed large increases in both the frequency and amplitude of miniature inhibitory postsynaptic currents (mIPSCs) (*Figure 2C*). Together, these findings indicate that the embryonic loss of MEF2C in cortical excitatory neurons results in a small decrease in glutamatergic synaptic transmission and a large increase in inhibitory synaptic transmission – the combination of which likely contributes to the dramatic reduction in cortical network activity as detected by spontaneous UP states.

## MEF2C regulates both excitatory and inhibitory synapses in a cell-autonomous manner

Since changes in UP states or the frequency of mEPSCs or mIPSCs can result from an alteration in synapse number, we next analyzed the density of excitatory and inhibitory synapses and overall dendritic complexity. While the *Mef2c* cKO cortical pyramidal neurons have a normal dendritic complexity (*Figure 3—figure supplement 1A,B*), we detected a ~2-fold increase in dendritic GABAergic synapse density on *Mef2c* cKO cortical pyramidal neurons (*Figure 3A*) as determined by quantifying co-localization of pre- and postsynaptic markers (GAD65 and GABRG2, respectively). We also detected a significant reduction in dendritic spine density on *Mef2c* cKO neurons (*Figure 3B* and *Figure 3—figure supplement 1E*), suggesting that MEF2C regulates, directly or indirectly, the densities of both excitatory and inhibitory synapses. The direction of these changes in E/I synapse density argues against the likelihood that the increase in inhibitory synapse density is a compensatory reaction to the reduction in excitatory synapse density, and vice versa, as these changes work in the same direction to reduce overall network synaptic activity.

To determine if the MEF2C-dependent synaptic changes are cell autonomous or represent a consequence of indirect network changes, we cultured *Mef2c^{fl/fl}* cortical neurons and generated a sparse population of individual *Mef2c* cKO neurons (transfection efficiency <1%) by transient transfection of Cre-recombinase or an enzyme-dead mutant form of Cre (△Cre). In this way, >99% of the presynaptic inputs onto the transfected neurons are wild-type, so noted changes in the synapse density onto Cre-expressing neurons are likely postsynaptic cell-dependent functions for MEF2C. Using this approach, we observe that Cre-expressing cortical pyramidal neurons displayed a significant increase in GABAergic synapse density (*Figure 3C*) and a significant decrease in dendritic spine density (*Figure 3D*), similar to the direction and magnitude of E/I synapse changes observed in the global *Mef2c* cKO neurons (*Figure 3A,B*). Interestingly, we did not observe any significant changes in GABAergic synapse density onto GAD65-positive *Mef2c^{fl/fl}* interneurons transfected with Cre (*Figure 3E*), suggesting the increase in inhibitory synapses is specific to excitatory cortical pyramidal neurons.

Overexpression of a transcription-promoting form of MEF2C (MEF2C-VP16) reduces structural and functional glutamatergic synapse density in hippocampal pyramidal neurons (*Flavell et al., 2006*; *Pfeiffer et al., 2010*; *Tsai et al., 2012*; *Zang et al., 2013*), similar to the effect of *Mef2c* loss-of-function mutation in cortical neurons. As such, we considered the possibility that MEF2C-VP16 and MEF2C loss-of-function could both produce excitatory synapse reductions if endogenous MEF2C is functioning as a transcriptional repressor on key target genes that regulate synapse density. Consistent with this basic idea, we found that overexpression of MEF2C-VP16, a constitutively-active form of MEF2C (*Figure 3—figure supplement 1F*), in wild-type cortical pyramidal neurons significantly reduced dendritic spine density (*Figure 3F*), whereas a dominant-repressor form of MEF2C (MEF2C-EN) (*Figure 3—figure supplement 1F*) failed to alter dendritic spine density (*Figure 3F,G*). However, cell-autonomous expression of MEF2C-EN in *Mef2c* cKO neurons rescued the decrease in dendritic spine density (*Figure 3G*), strongly suggesting that endogenous MEF2C functions as a transcriptional repressor to regulate dendritic spine density. In wild-type cortical pyramidal neurons, MEF2C-VP16 also significantly increased GABAergic synapse density (*Figure 3H*), whereas MEF2C-EN had no effect on GABAergic synapse density (*Figure 3H,I*). Similar to the dendritic spine density findings, cell-autonomous expression of MEF2C-EN rescued the increase in GABAergic synapse density observed in the *Mef2c* cKO cortical pyramidal neurons (*Figure 3I*). These results suggest that endogenous MEF2C functions predominantly as a transcriptional repressor to inhibit target genes that promote excitatory synapse elimination and inhibitory synapse formation and/or stability.

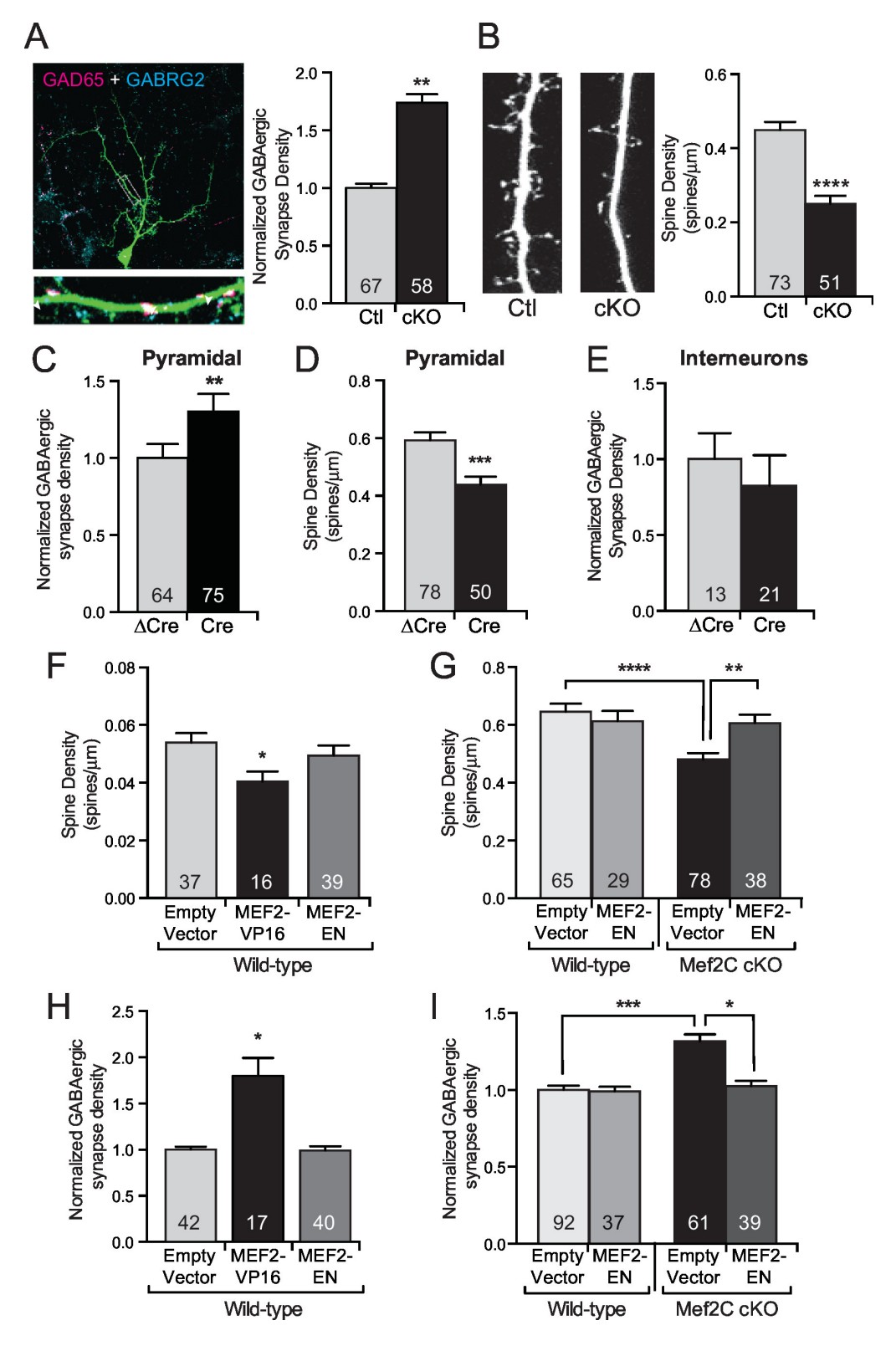

**Figure 3.** MEF2C functions as a transcriptional repressor to regulate synapse development in postsynaptic cortical pyramidal neurons. (**A**) Representative image of a GFP expressing mouse cortical neuron immunostained with antibodies against GAD65 (pre-synaptic) and GABRG2 (post-synaptic). Quantification of inhibitory synapse density (see Materials and methods) on *Mef2c* cKO neurons showed an increase compared to wildtype control neurons. (**B**) Representative image of spine density across a dendritic stretch. Quantification of spine density on *Mef2c* cKO neurons showed a

*Figure 3 continued*

reduction compared to wildtype control neurons. (**C**) Quantified GABAergic synapse density onto *Mef2c^fl/fl*cortical pyramidal neurons transfected at DIV4 with either Cre-GFP (Cre) or an enzyme-dead mutant of Cre-GFP (ΔCre). (**D**) Quantified spine density onto *Mef2c^fl/fl* cortical pyramidal neurons transfected at DIV4 with either Cre-GFP (Cre) or an enzyme-dead mutant of Cre-GFP (ΔCre). (**E**) Quantified GABAergic synapse density onto *Mef2c^fl/fl*GAD65 positive interneurons transfected at DIV4 with either Cre-GFP (Cre) or an enzyme-dead mutant of Cre-GFP (ΔCre). (**F**) Quantified spine density onto WT cortical pyramidal neurons transfected at DIV4 with either an empty vector, a constitutive transcriptional promoting form of MEF2C (MEF2-VP16), or a constitutive transcription repressor form of MEF2C (MEF2-EN). (**G**) Quantified spine density onto wildtype or *Mef2c* cKO neurons transfected with either an empty vector or MEF2-EN. (**H**) Quantified GABAergic synapse density onto WT cortical pyramidal neurons transfected at DIV4 with an empty vector, a constitutive transcriptional promoting form of MEF2C (MEF2-VP16), or a constitutive transcription repressor form of MEF2C (MEF2-EN). (**I**) Quantified GABAergic synapse density onto wildtype or *Mef2c* cKO neurons transfected with an empty vector or MEF2-EN. Data are represented as mean ± SEM. Number (n) of neurons (A,C,E,H,I) or of dendritic stretches (B,D,F,G) are reported in each bar. Statistical significance was determined by unpaired t-test (**A–E**), One-way ANOVA (**F,H**) or Two-way ANOVA (**G,I**) using GraphPad Prism. *p<0.05, **p<0.01, ***p<0.001, ****p<0.0001. Also see *Figure 3—figure supplement 1*.

The following figure supplement is available for figure 3:

**Figure supplement 1.** Structural synaptic changes in *Mef2c* cKO neurons.

## Differential gene expression in *Mef2c* cKO cortex

Since MEF2C is a nuclear transcription factor, we sought to identify differential gene expression that results from early embryonic deletion of *Mef2c*. To this end, we performed deep sequencing of polyA-enriched mRNAs (RNA-Seq) isolated from the somatosensory cortex of *Mef2c* cKO or control littermates and identified differentially expressed genes (DEGs) (*Figure 4A*; *Supplementary file 1*). Using a stringent cut-off (|log2FC| > 0.3, FDR < 0.05), we detected 1076 DEGs in *Mef2c* cKO cortex compared with controls, including 598 genes with a decreased expression level and 478 genes with an increased expression level. Comparing the *Mef2c* cKO DEGs to genes expressed in specific neuronal populations (*Cahoy et al., 2008*), the DEGs were highly enriched for neuron-expressed genes (*Figure 4—figure supplement 1A,B*), consistent with the observation that *Mef2c* is not expressed at appreciable levels in non-neuronal cells (*Figure 1A*). Notably, the expression levels of *Mef2a* and *Mef2d*, which are close family members, were not significantly altered in the *Mef2c* cKO mice (*Figure 4C*).

Gene ontology enrichment analysis revealed multiple distinct categories for up-regulated and down-regulated genes (*Figure 4—figure supplement 1C*; *Table 1*). For instance, genes with an increased expression level in the young adult *Mef2c* cKO cortex were significantly enriched for cellular processes such as neuron differentiation and development, suggesting that MEF2C might function as a repressor for some of these DEGs. Analysis of down-regulated genes in *Mef2c* cKO mice revealed a significant enrichment for cellular processes including synaptic transmission and ion transport (p<0.01, BH correction; *Figure 4—figure supplement 1C*), suggesting an important role for MEF2C in regulating synapse function and neuronal excitability.

To further characterize the *Mef2c* cKO DEGs, we compared our gene list with the recently updated risk genes from the Simons Foundation Autism Research Initiative (SFARI database, 667 genes) (*Basu et al., 2009*), mRNAs associated with FMRP (*Darnell et al., 2011*), ID-associated genes from multiple sources (*Inlow and Restifo, 2004*; *Lubs et al., 2012*; *Ropers, 2008*; *van Bokhoven, 2011*), and synaptic-associated genes (Synaptome DB) (*Pirooznia et al., 2012*). In the *Mef2c* cKO DEGs, we identified a significant overrepresentation of ASD-risk genes (p=0.0007, hypergeometric test, perm=0.001) and synapse-linked genes (p=0.0004, hypergeometric test, perm=0.001), (*Figure 4B*), including the autism-linked genes *Ntng1, Nlgn1, Nrxn1, Nrxn3, Pcdh19, Shank2, Shank3, Pten* and *Htr1b*. We also detected a significant enrichment for FMRP-associated RNAs (p=$3\times10^{-07}$, hypergeometric test, perm=0.001), which is interesting since FMRP is required for MEF2C-VP16-induced excitatory synapse elimination (*Pfeiffer et al., 2010*). Using quantitative PCR (qPCR), we validated dysregulation of several *Mef2c* cKO DEGs in both postnatal day 21 (P21) and adult SSC tissue (*Figure 4C,D*). Interestingly, there were several dysregulated genes that are reported to regulate inhibitory GABAergic transmission (*Gabra5* and *Nos1*), and we observed a significant increase in *Pcdh10* mRNA, a factor we previously implicated in MEF2/FMRP-dependent glutamatergic synapse elimination (*Tsai et al., 2012*). Overall, our RNAseq data analysis suggests that

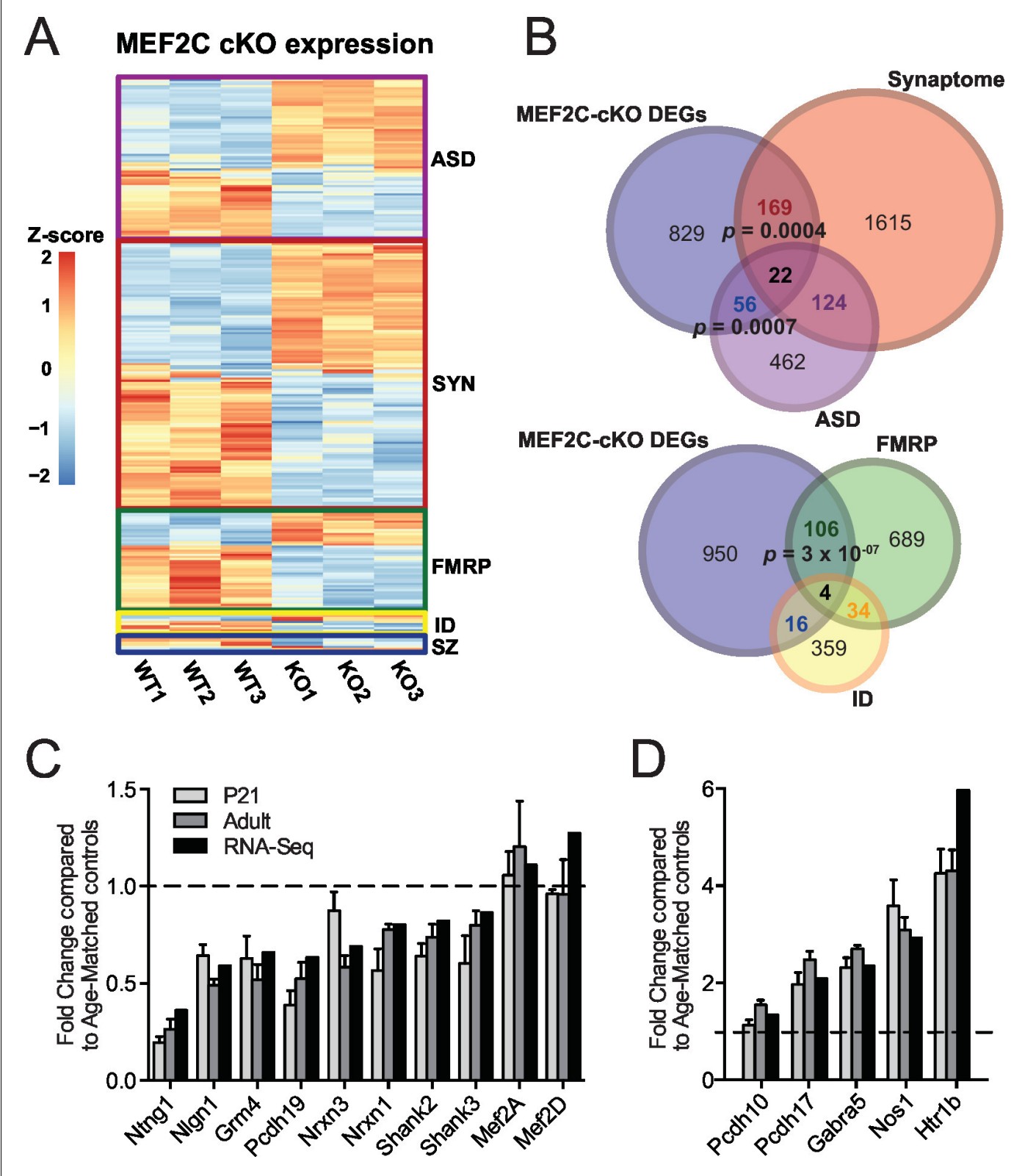

**Figure 4.** Characterization of *Mef2c* cKO RNA-Seq differentially expressed genes. (A) Heatmap showing the disorder-related genes differentially expressed in *Mef2c* cKO (KO) compared with wild-type (WT). In red, genes with higher expression; in blue, genes with lower expression. (B) Overlap between *Mef2c* cKO DEGs and gene sets of interest. Marked, the overlap p-values. Number of genes for each gene sets are indicated. (C) Relative expression of selective down-regulated ASD-associated genes from *Mef2c* cKO DEGs compared to controls. Both RNA-Seq and qPCR (P21 and Adult)

*Figure 4 continued on next page*

*Figure 4 continued*

show similar expression changes for most genes. (D) Relative expression of selective up-regulated *Mef2c* cKO DEGs compared to controls. Both RNA-Seq and qPCR (P21 and Adult) show similar expression changes. Data are represented as mean ± SEM. See Materials and methods for statistical analysis. n=3 animals/genotype for RNA-Seq; n=6 animals/genotype for adult qPCR; n=6 control and n=4 *Mef2c* cKO for P21 qPCR. Also see *Table 1*, *Supplementary file 1* and *Figure 4—figure supplement 1*.

The following figure supplement is available for figure 4:

**Figure supplement 1.** Differential gene expression in *Mef2c* cKO cortical tissue.

MEF2C, either directly or indirectly, influences a large, complex gene expression program that influences neuronal and synapse development, and numerous syndromic and idiopathic autism-linked genes.

## *Mef2c* cKO mice display behavioral phenotypes relevant to neurodevelopmental disorders

Since we observed significant dysregulation of many ASD-related genes in *Mef2c* cKO mice, and since *MEF2C* is linked to human neurodevelopment disorders with autistic features and cognitive deficits, we examined whether loss of MEF2C function in forebrain excitatory neurons might produce autism- and ID-like behavioral phenotypes. In humans, impairments in communication and social interactions are common symptom domains of autism and SCZ. *Mef2c* cKO mice displayed dramatic abnormalities in a putative form of oral social communication in mice – ultrasonic vocalizations (USVs) produced by a young adult male mouse when placed in the presence of a female in estrous or upon young pup separation from its mother (*Figure 5*) (*Ey et al., 2013*; *Hanson and Hurley, 2012*). In the presence of a sexually-receptive female, *Mef2c* cKO males generated far fewer USV calls (~70% reduction, *Figure 5B*), and showed a significant increase in the latency to the first call

**Table 1.** Gene-ontology of *Mef2c* cKO DEGs.

| Category | Term | Count | Benjamini | log |
|---|---|---|---|---|
| UP | neuron projection development | 22 | 2.49E-05 | 46.03800653 |
| UP | neuron development | 24 | 7.33E-05 | 41.34896025 |
| UP | axonogenesis | 18 | 9.25E-05 | 40.33858267 |
| UP | neuron projection morphogenesis | 18 | 0.00014 | 38.53871964 |
| UP | cell projection organization | 24 | 0.000175 | 37.56961951 |
| UP | cell morphogenesis involved in neuron differentiation | 18 | 0.000181 | 37.42321425 |
| UP | cell projection morphogenesis | 18 | 0.00055 | 32.59637311 |
| UP | neuron differentiation | 26 | 0.000569 | 32.44887734 |
| UP | cell morphogenesis involved in differentiation | 18 | 0.000822 | 30.85128182 |
| UP | cell part morphogenesis | 18 | 0.000822 | 30.85128182 |
| UP | cell morphogenesis | 20 | 0.008075242 | 20.92844454 |
| UP | cellular component morphogenesis | 20 | 0.03538264 | 14.51209766 |
| DOWN | potassium ion transport | 28 | 7.00E-09 | −8.15E+01 |
| DOWN | metal ion transport | 44 | 5.62E-10 | −92.50263684 |
| DOWN | cation transport | 45 | 1.63E-08 | 7.79E+01 |
| DOWN | ion transport | 50 | 1.25E-06 | −59.03089987 |
| DOWN | synaptic transmission | 21 | 3.13E-05 | −45.04455662 |
| DOWN | transmission of nerve impulse | 23 | 7.95E-05 | −40.99632871 |
| DOWN | cell-cell signaling | 25 | 0.00035 | −34.55931956 |

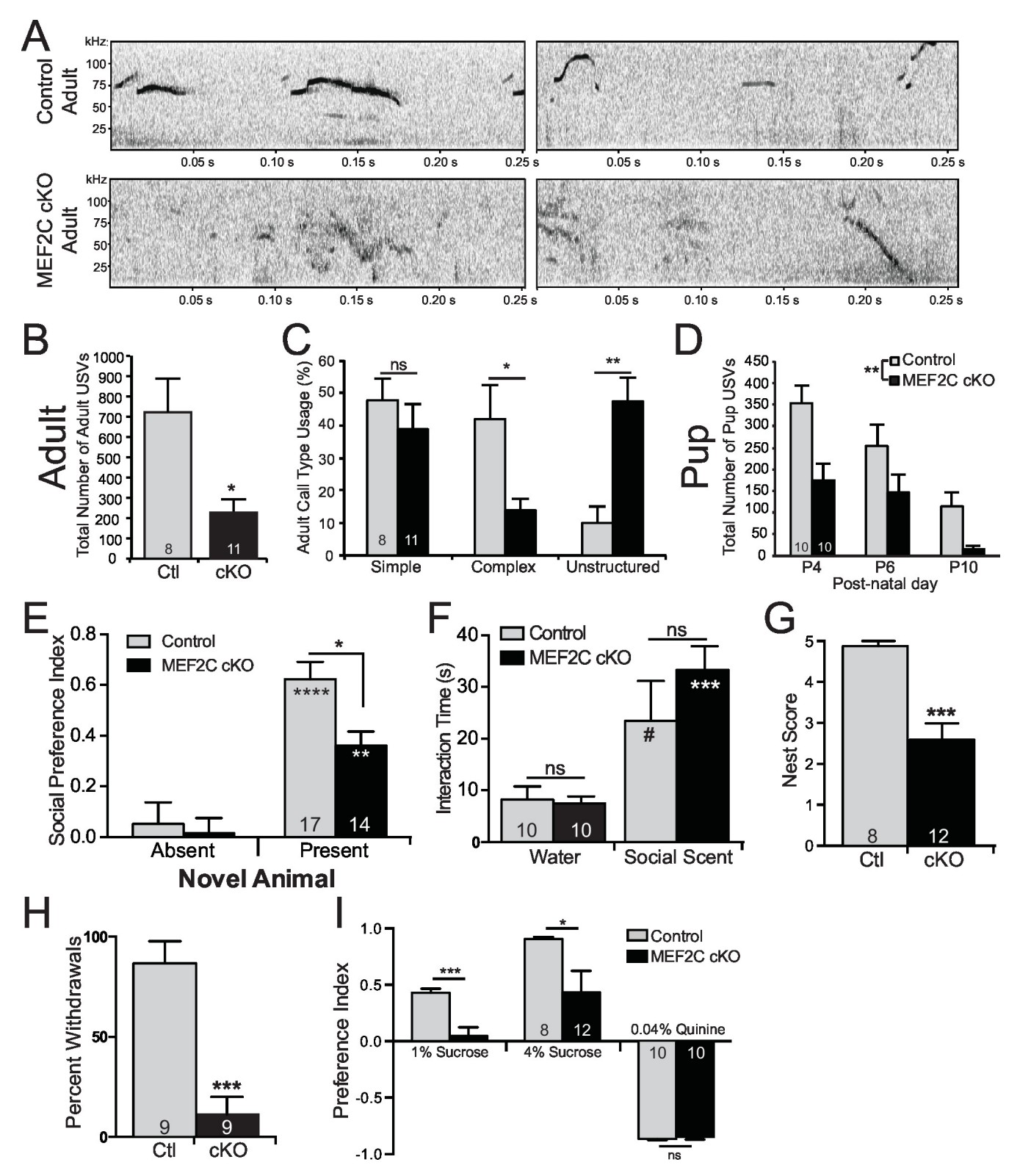

**Figure 5.** Social behavior abnormalities in *Mef2c* cKO mice. (**A**) Representative spectrograms of ultrasonic vocalizations (USVs) recorded from adult male mice in the presence of an estrous female mouse. (**B**) Adult *Mef2c* cKO male mice emit fewer USVs to an estrous female than control littermates. (**C**) Adult *Mef2c* cKO male mice show different call types than control littermates. *Mef2c* cKO mice have more unstructured USVs (%) and fewer complex USVs than control mice. Representative images of call types and further breakdown of USV sub-type are presented in *Figure 5—figure supplement*

*Figure 5 continued on next page*

Figure 5 continued

*1A,B*. (D) Juvenile *Mef2c* cKO mice (pups) emit fewer USVs during maternal separation than control littermates. USVs were recorded on postnatal days (P) 4, 6, and 10. (E) *Mef2c* cKO mice show reduced preference for interacting with a novel social target. (F) *Mef2c* cKO mice show normal olfactory response to novel social scent. (G) *Mef2c* cKO mice fail to build structured nest when utilizing a nest score system (*Deacon, 2006*). (H) *Mef2c* cKO mice induce control littermates to withdrawal in the tube test for social dominance in >90% of trials. (I) *Mef2c* cKO mice show reduced preference for a natural reward, sucrose. Both genotypes showed aversion to the bitter solution, 0.04% quinine. Data are represented as mean ± SEM. Statistical significance was determined by unpaired t-test (B,G,H–I) or 2-way ANOVA with Sidak's post-hoc comparison (C–F). #p<0.1, *p<0.05, **p<0.005, ***p<0.0005, ns=not significant. Numbers of animals (n) are reported in each bar for respective experiment. Also see *Figure 5—figure supplement 1*.

The following figure supplement is available for figure 5:

**Figure supplement 1.** Characterization of *Mef2c* cKO mouse USVs.

(~3-fold, *Figure 5—figure supplement 1C*). Wild-type littermate mice produced a range of distinct simple and complex stereotyped USV call subtypes (*Figure 5—figure supplement 1A*) (*Ey et al., 2013*). In contrast, *Mef2c* cKO mice produced a ~5-fold increase in unstructured USVs, and a corresponding decrease in complex, but not simple, USVs (*Figure 5C* and *Figure 5—figure supplement 1B*). While many basic USV parameters were unchanged by genotype (*Figure 5—figure supplement 1D–F*), we detected significant reductions in the maximum USV frequency and the mean frequency at the end of calls (*Figure 5—figure supplement 1G–H*). Similar to the adults, *Mef2c* cKO mice at postnatal days 4–10 produced significantly fewer USVs upon separation from the mother (distress calls) (*Figure 5D*), but at this age, the USV structures, subtypes and basic call parameters were indistinguishable from WT littermates (*Figure 5—figure supplement 1I–J*). Together these data indicate that *Mef2c* cKO mice produce significantly fewer USVs in a species-specific form of putative oral communication.

In a social interaction test, the wild-type littermates spent significantly more time interacting with an unfamiliar mouse than an empty chamber (*Figure 5E*). The *Mef2c* cKO mice also spend more time interacting with the social animal vs. the empty chamber, but the *Mef2c* cKO mice spent significantly less time interacting with the social animal than the control mice (*Figure 5E*). In another cohort of mice, we observed a significant reduction in social interaction even in the presence of a competing novel inanimate object (*Figure 5—figure supplement 1K*). The reduction in social interaction did not appear to be due to deficits in olfactory recognition of social animals or basic novelty detection, since *Mef2c* cKO mice showed a strong preference to interact with a social-related smell from an unfamiliar mouse (*Figure 5F*). In addition to social interaction deficits, *Mef2c* cKO mice showed significant reductions in another social-related behavior, nest building (*Figure 5G*) (*Etherton et al., 2009*; *Kwon et al., 2006*; *Deacon, 2006*), and they displayed abnormal social behavior in the tube test for social dominance (*Figure 5H*).

Deficits in brain reward function have been proposed to contribute to some autistic behaviors, including social interaction (*Insel, 2003*; *Dichter et al., 2012*), and a lack of motivation is a common negative symptom of SCZ. Interestingly, *Mef2c* cKO mice show significant reductions in a hedonic-related behavior in the sucrose preference test (*Figure 5I*), an assay that measures an animal's preference for a sweet solution vs. water. The reduction in sucrose preference is not likely due to basic gustatory deficits since *Mef2c* cKO mice showed normal avoidance of a bitter-tasting solution (quinine) (*Figure 5I*, right). Taken together, *Mef2c* cKO mice demonstrate multiple abnormalities in mouse social behaviors and a strong deficit in an appetitive reward-related behavior.

Autism is characterized by restricted or repetitive patterns of behavior, interests or activities (*American Psychiatric Association, 2013*). Interestingly, *Mef2c* cKO mice spent a significantly greater fraction of time in a repetitive jumping behavior (~3-fold increase, *Figure 6A*), which was visually observed in both novel and home cage settings. In addition, we detected a significant increase in repetitive fine motor movements (*Figure 6B*), which is often interpreted as a motor stereotypy behavior (*Avale et al., 2004*). In contrast, no differences by genotype were observed in time spent self-grooming or digging (*Figure 6—figure supplement 1A,B*). *Mef2c* cKO displayed normal motor coordination as measured in the accelerating rotarod test (*Figure 6C*), but significant motor hyperactivity was detected in a novel environment (*Figure 6D*).

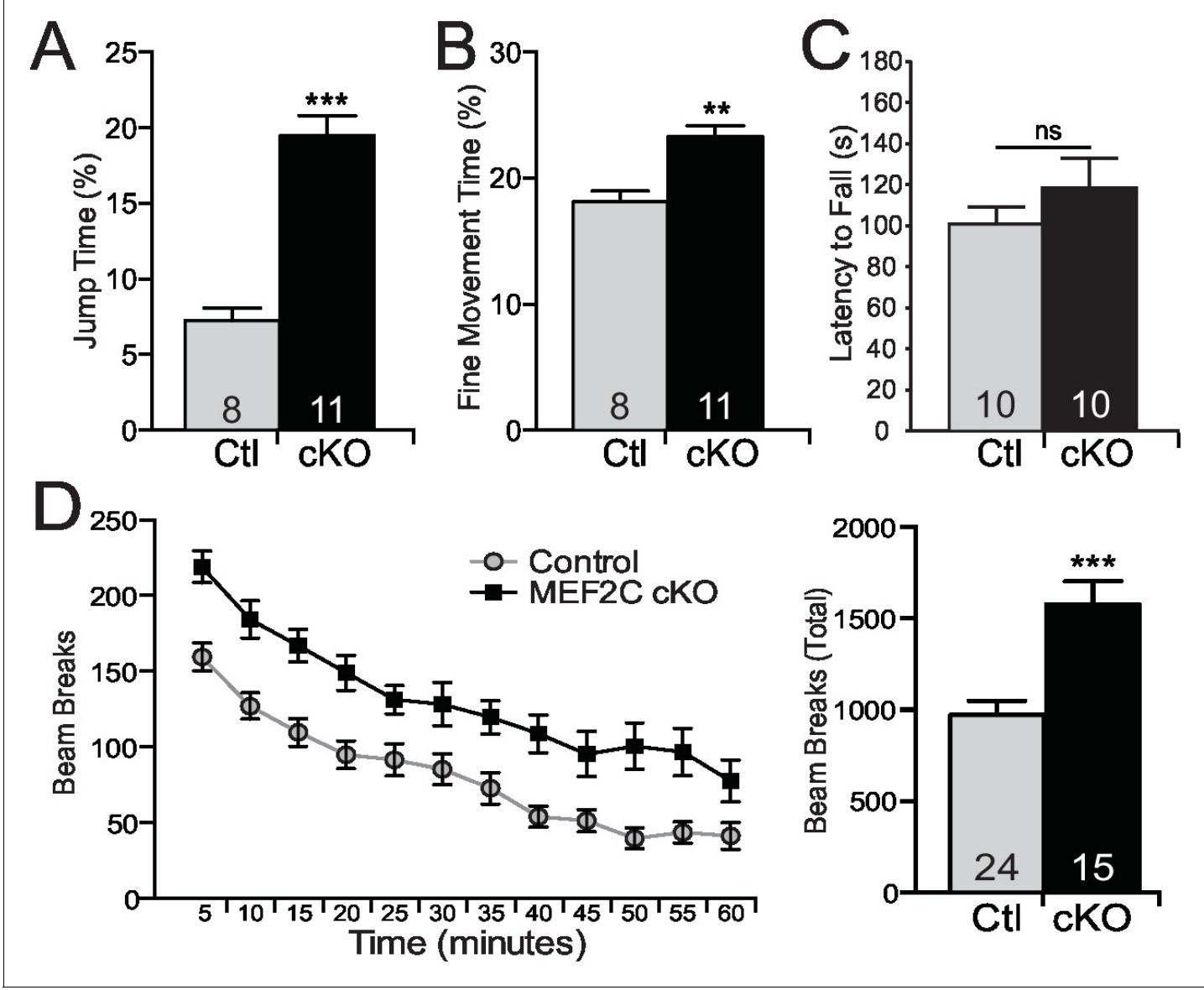

**Figure 6.** Repetitive behaviors and hyperactivity in *Mef2c* cKO mice. (A) *Mef2c* cKO mice spend more time jumping than control animals in an operant chamber over a 1-hr interval. (B) *Mef2c* cKO mice have more fine motor movements in an operant chamber, reflective of stereotypic activity. (C) Latency to fall off an accelerating rotarod is not different in the *Mef2c* cKO mice. (D) *Mef2c* cKO mice are hyperactive compared to control littermates. Activity was monitored for 1 hr, and data is plotted by beam breaks/5 min (left) and cumulative beam breaks (right). Data are represented as mean ± SEM. Statistical significance was determined by unpaired t-test (A–D) or 2-way ANOVA (D). **p<0.005, ***p<0.0005, ns=not significant. Numbers of animals (n) are reported in each bar for respective experiment.
The following figure supplement is available for figure 6:

**Figure supplement 1.** *Mef2c* cKO mice do not exhibit repetitive grooming or digging.

Intellectual disability is also a common associated symptom of autism (*American Psychiatric Association, 2013*), and cognitive deficits comprise one of the three major symptom domains in SCZ. *Mef2c* cKO and control littermates showed similar startle responses to a broad range of foot shock intensities and acoustic white-noise volumes (*Figure 7A,B*), suggesting that nociception and auditory sensory sensitivity are not significantly altered in the mutant mice. However, the *Mef2c* cKO mice showed profound deficits in threat-related learning and memory in the classic fear-conditioning

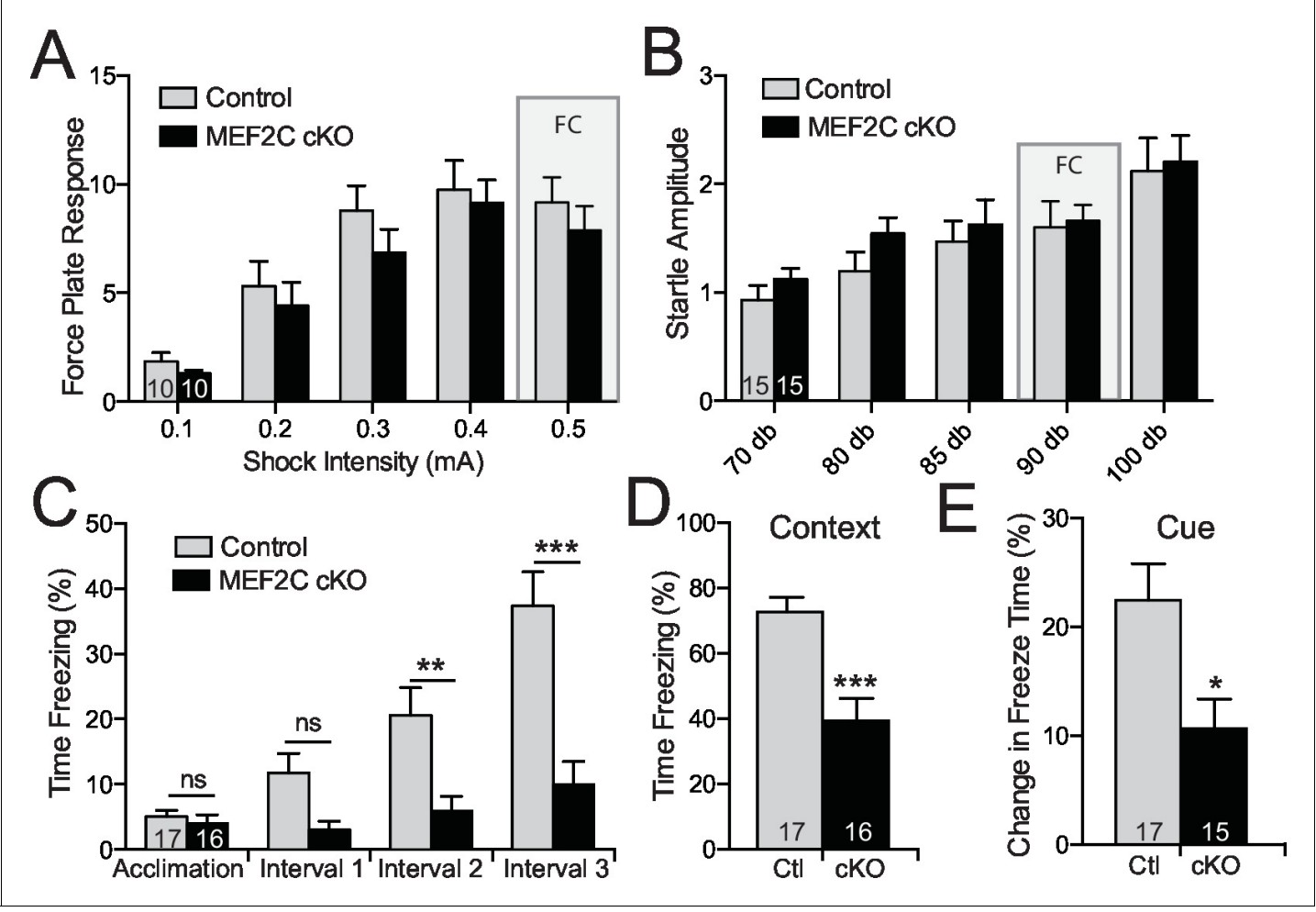

**Figure 7.** Cognitive deficits in *Mef2c* cKO mice. (A–B) Both control and *Mef2c* cKO mice showed similar force plate response to various intensities of shock (A) or acoustic startle (B). Grey bars highlight the intensities used in fear conditioning (FC). (C) During training for fear conditioning, *Mef2c* cKO mice fail to increase freezing during the 1-minute intervals after each tone/shock pairing. (D) Fear Conditioning. *Mef2c* cKO mice show deficits in contextual memory. (E) Fear Conditioning. In a novel context, *Mef2c* cKO mice show deficits in cue-dependent memory. Data are represented as mean ± SEM. Statistical significance was determined by 2-way ANOVA (A–C) or unpaired t-test (D,E). *p<0.05, **p<0.005, ***p<0.0005, ns=not significant. Numbers of animals (n) are reported in each bar for respective experiment.

assay (*Figure 7C–E*). Unlike the WT controls, the *Mef2c* cKO mice failed to develop robust freezing behaviors in the 1 min. periods following each tone-shock pairing (*Figure 7C*). Twenty-four hours later, *Mef2c* cKO mice showed significantly reduced freezing behaviors when re-exposed to the shock-paired context (*Figure 7D*) or after presentation of the tone cue (conditioned stimulus) in an altered context (*Figure 7E*). Together these findings suggest that embryonic loss of MEF2C in excitatory forebrain neurons causes significant deficits in fear learning and memory, multiple social behaviors, socially-motivated ultrasonic vocalizations, and reward-related behaviors. *Mef2c* cKO mice also show significant increases in repetitive motor behaviors and overall hyperactivity – all symptom domains with potential relevance to human neurodevelopmental disorders such as autism, ID and SCZ.

# Discussion

We conditionally deleted *Mef2c* in Emx1-lineage populations, including cortical excitatory neurons, during early embryogenesis, and observed a large reduction in cortical network activity (UP states), a small decrease in layer 2/3 pyramidal neuron excitatory synaptic transmission and a large increase in inhibitory synaptic transmission in primary sensory cortex. Consistent with these ex vivo acute slice recordings, cultured *Mef2c* cKO cortical neurons displayed a significant decrease in dendritic spine density and an increase in GABAergic synapse density. Our functional rescue findings suggest that MEF2C in cortical pyramidal neurons regulates E/I synapse densities in early development by acting in the postsynaptic neuron as a cell-autonomous, transcriptional repressor on key target genes. Loss of MEF2C alters the gene expression, directly or indirectly, of numerous autism- and synapse-linked genes and RNAs known to associate with FMRP, the leading genetic cause of ID and autism. These findings position MEF2C as a critical transcriptional regulator functioning at the nexus of numerous synapse- and neurodevelopmental disorder-linked genes. Finally, the *Mef2c* cKO mice display multiple behavioral phenotypes that are reminiscent of core autism symptoms in humans, including abnormal social behaviors, reduced USVs in multiple ages and contexts, and an increased frequency of some repetitive motor behaviors. We also observed learning and memory deficits and motor hyperactivity, which are common associated symptoms of ASDs and observed in patients with *MEF2C* mutations. Taken together, our findings suggest that the loss of MEF2C's repressor function in cortical excitatory neurons produces an E/I synapse imbalance and that this synaptic phenotype might contribute to the numerous, neurodevelopmental disorder-related behavioral phenotypes observed in the *Mef2c* cKO mice.

Previous studies have demonstrated an important role for MEF2A and MEF2D in the process of activity-dependent excitatory synapse elimination in hippocampal neurons (*Flavell et al., 2006*; *Pfeiffer et al., 2010*; *Tsai et al., 2012*). However, similar to a previous report (*Li et al., 2008*), we found that loss of MEF2C produced a *decrease* in cortical excitatory synaptic transmission, suggesting that MEF2C is a positive regulator of excitatory synapses in cortical neurons. However, overexpression of MEF2C-VP16, a strong, constitutive transcriptional activator form of MEF2C, in wild-type cortical neurons also decreased dendritic spine density and increased inhibitory synapse density (*Figure 3F,H*) – essentially phenocopying the *Mef2c* cKO neurons. In contrast, overexpression of MEF2C-EN, a dominant repressor form of MEF2C, in WT neurons had no effect on dendritic spines or GABAergic synapse densities; whereas, expression of MEF2C-EN in *Mef2c* cKO cortical pyramidal neurons normalized excitatory and inhibitory synapse densities to WT levels (*Figure 3G,I*). Together, these findings suggest that endogenous MEF2C functions as a cell-autonomous, transcriptional repressor at one or more key target genes involved in excitatory synapse elimination and GABAergic synapse formation/stability. These observations are consistent with previous reports demonstrating that MEF2A can also function as a transcriptional repressor to promote cerebellar excitatory synapse maturation (*Shalizi et al., 2006*), highlighting a critical role for diverse mechanisms of transcriptional regulation by MEF2 family members to regulate synapse development in the brain. It's interesting to note that membrane depolarization of *Mef2c* cKO cortical neurons stimulated MEF2 reporter gene activity levels that are comparable to responses in WT littermates, suggesting that the activity-dependent induction of MEF2-sensitive target genes can be fully compensated for by endogenous MEF2A and MEF2D. In the future, it will be important to study the overlapping and potentially distinct functions of the various MEF2 family members, and their activity-dependent regulation, in cortical neuron function and development.

Loss of MEF2C in forebrain excitatory neurons produces an *increase* in structural and functional excitatory synapses formed onto hippocampal dentate granule neurons (DG) (*Barbosa et al., 2008*; *Adachi et al., 2016*), suggesting that MEF2C might have cell-type specific functions and/or that the increase in DG excitatory synaptic transmission is an indirect, homeostatic effect of decreased cortical stimulation of DG neurons. In the future, cell autonomous manipulations of the DG neurons will be important to resolve this question. Also, postnatal *Mef2c* deletion in forebrain excitatory neurons did not produce social or repetitive behavioral phenotypes, despite the increase in DG dendritic spine density (*Adachi et al., 2016*), suggesting a dissociation of hippocampal DG spine density and postnatal MEF2C deletion from several ASD-related behaviors. As such, the role(s) for MEF2C in embryonic and/or early postnatal cortical development might be more critical for producing the behavioral phenotypes observed in our *Mef2c* cKO mice. Finally, while we observe significant

synaptic and gene expression differences in the SSC of the *Mef2c* cKO mice, Emx1-Cre expression is also detected in other brain regions and cell types, including the olfactory bulb and some glia, and as such, it is not possible to attribute the behavioral phenotypes observed in the *Mef2c* cKO mice to the synaptic changes in the SSC, and future studies that further dissect the Emx1-lineage populations will be necessary to link specific *Mef2c* cKO behavioral phenotypes with a specific brain region(s).

Imbalances in excitatory and inhibitory synaptic transmission are proposed to underlie many neuropsychiatric disorders, including ASDs (*Rubenstein, 2010*; *Cellot and Cherubini, 2014*) and SCZ (*Coyle et al., 2016*). Genetic analyses of patients affected by these disorders revealed mutations in many synapse-related genes (*Garber, 2007*; *McCarthy et al., 2014*). In mice, *increased* excitatory synaptic function has been reported in several mouse ASD models, including mutant mice lacking *Fmr1*, *Pten*, and *Tsc1/2* genes (*Gibson et al., 2008*; *Williams et al., 2015*; *Bateup et al., 2013*). In contrast, only a few prior studies have examined the role of altered inhibitory synapse function in ASD-related behaviors. For example, mice containing a human disease mutation in the *Nlgn3* gene (*Nlgn3 R451C*) displayed an increase in inhibitory synaptic transmission and several autism-associated behaviors (*Tabuchi et al., 2007*), suggesting that altered E/I balance in either direction can produce behavioral phenotypes with potential relevance to neurodevelopmental disorders.

Little is currently known about the mechanisms that control GABAergic synapse density (*Petrini and Barberis, 2014*). Interestingly, GABAergic synapse formation during development precedes glutamatergic synapse formation (*Ben-Ari et al., 2007*), and it can strongly affect the subsequent development of glutamatergic synapses and neuronal morphology (*Wang and Kriegstein, 2008*). Our finding here show that MEF2C in postsynaptic cortical neurons function to cell-autonomously modulate the density of GABAergic synapses formed on the dendrites of cortical pyramidal neurons. In contrast, loss of MEF2C on cortical interneurons did not alter GABAergic synapse density (*Figure 3E*), suggesting a specific role in cortical pyramidal neurons. Future studies will focus on exploring the precise regulation, molecular mechanisms and developmental period(s) when MEF2C regulates inhibitory and/or excitatory synapse density.

Comparison of differentially expressed genes (DEGs) in the SSC of *Mef2c* cKO mice by RNA-Seq revealed ~1000 significantly altered genes. Nearly half of the dysregulated genes in the *Mef2c* cKO cortex showed an increased expression level compared to controls, consistent with the idea that MEF2C can function as a transcriptional repressor, directly or indirectly, on a subset of these DEGs. Among all the dysregulated genes, we observed significant overlap of *Mef2c* cKO DEGs and autism-linked genes (78), synaptome genes (191), and FMRP-target RNAs (110), many of which are associated with multiple disease and neuronal function groups (*Figures 4B* and *Figure 4—figure supplement 1C*), including *Nlgn1* (ASD, Synaptome), *Nrxn1/3* (ASD, Synaptome, FMRP-bound RNAs), *Grm4* (ASD, FMRP-bound RNAs), and *Shank2/3* (ASD, Synaptome, FMRP-bound RNAs). Additionally, we found reduced *Cdkl5* expression in the *MEF2C* cKO DEGs, consistent with the reduced expression of *CDKL5* mRNA reported in human patients with *MEF2C* mutations (*Zweier et al., 2010*).

Recent human genetic studies have revealed that deletion of, or non-synonymous mutations in, the *MEF2C* gene is associated with a severe neurodevelopmental disorder with features of autism and ID (*Paciorkowski et al., 2013*; *Mikhail et al., 2011*; *Novara et al., 2010*; *Le Meur et al., 2010*; *Cardoso et al., 2009*; *Engels et al., 2009*). These *MEF2C* haploinsufficient patients present with a number of symptoms, including motor abnormalities (*e.g.* dyskinesias, stereotypies and hyperactivity), impairments in reciprocity, severe deficits in verbal communication, and severe intellectual disability (*Paciorkowski et al., 2013*). In humans, *MEF2C* haploinsufficiency appears to be sufficient, at least in reported individuals, to produce this complex and severe neurodevelopmental disorder. Generally consistent with previous reports (*Li et al., 2008*; *Barbosa et al., 2008*), our preliminary studies indicate that loss of one gene copy of *Mef2c* (*Mef2c$^{fl/+}$;Emx1$^{Cre}$/$^{+}$*) in the Emx1-cell lineage produces mice with behaviors indistinguishable from their Cre-negative WT controls (A.J.H. and C. W.C., unpublished observations). It is possible that in humans there are other factors that influence disease penetrance and severity, including unique or sensitized functions for MEF2C through human evolution, human-specific genetic modifiers and/or environmental influences that increase symptom penetrance or additional cell populations where reduction in MEF2C is required. Nevertheless, our findings here indicate that MEF2C plays an essential role in early cortical synaptic development, and

that reduction in MEF2C function in forebrain excitatory neurons can produce behaviors potentially relevant to multiple intellectual and developmental disorders.

Schizophrenia is a debilitating mental illness with neurodevelopmental origins that affects nearly 1% of the world's population, and there is significant overlap in risk genes for ASDs and SCZ. In contrast to ASDs, human postmortem brain analysis of SCZ brains revealed a thinning of the cortex and a decrease in dendritic spine density, observations supporting the leading hypothesis that hypofunction of excitatory synaptic transmission underlies the pathophysiology of SCZ (*Coyle et al., 2016*). Recently, 108 genomic loci were identified by SNP meta-analysis as conferring significant risk for SCZ, and *MEF2C* was identified as a candidate risk gene (*Schizophrenia Working Group of the Psychiatric Genomics Consortium, 2014*). In the *Mef2c* cKO mice, we observed a thinning of the cortex (*Figure 1C*), a decrease in dendritic spine density of *Mef2c* cKO cortical neurons (*Figure 3B*), and behavioral phenotypes that are reminiscent of cognitive and negative symptoms of SCZ (*e.g.* learning and memory deficits, lack of pleasure and motivation, reduced sociability and poverty of speech). While the potential relevance of *Mef2c* cKO phenotypes to the pathophysiology and symptoms of ASDs, ID and/or SCZ is not yet clear, our findings reveal an essential role for MEF2C in cortical neuron development and typical animal behaviors.

In summary, we show here that *Mef2c* is required for proper synapse development on excitatory forebrain neurons, and its embryonic loss in these populations produces mice with behavior phenotypes reminiscent of multiple neurodevelopmental disorders, including ASDs and ID. The behavior changes are associated with a reduction in cortical network activity and alterations in E/I synapse densities and function. We also show that MEF2C likely regulates E/I synapse density by functioning as a cell-autonomous, transcriptional repressor, and that *Mef2c* loss-of-function in Emx1-lineage populations produces, directly or indirectly, a dramatic dysregulation of hundreds of neuronal genes, positioning it at the nexus of numerous critical neurodevelopment genes reported to influence neuronal and synaptic development and risk for neurodevelopmental disorders.

## Materials and methods

### Animals

Mice (*mus musculus*) were group housed (2–5 mice/cage; unless specified) with same-sex littermates on a 12 hr light-dark cycle with access to food and water ad libitum. *MEF2C^{fl/fl}* mice were previously described (*Arnold et al., 2007*), as were *Emx1^{Cre/+}* knock-in mice (*Iwasato et al., 2008*). Test mice were bred and maintained on a mixed SVeV-129/C57BL/6J background. Experimental mice (*Mef2c^{fl/fl}; Emx1^{Cre/+}*) were compared to Cre-negative littermates (*Mef2c^{fl/fl}*). Experimenters were blinded to the mouse genotype during data acquisition and analysis. Every experiment was independently replicated at least twice, and total numbers of animals/cells are reported in the respective figures. All procedures were conducted in accordance with the Institutional Animal Care and Use Committee (IACUC) and National Institute of Health guidelines.

### Data acquisition

All experiments were independently replicated at least twice (typically 3–4 times). The numbers of animals/neurons/dendritic stretches are reported in each figure, and these numbers were estimated based on previous reports. Outliers were determined using GraphPad's Outlier calculator and excluded from data analysis.

### Immunohistochemistry

Mice were terminally anesthetized with sodium pentobarbital (Sigma) and perfused transcardially with PBS followed by 4% (w/v) paraformaldehyde (PFA). Brains were post-fixed overnight at 4°C in 4% PFA then cryoprotected in 30% sucrose. Brains were coronally sectioned at 30 μm using a sliding microtome and stored in PBS with 0.02% sodium azide. Sections were blocked in 3% albumin from bovine serum (BSA) and 3% normal donkey serum in PBS with TritonX-100 (0.3%) and Tween20 (0.2%) for 2 hr at room temperature (RT). Sections were immunostained overnight at 4°C anti-NeuN (A60, 1:200; Millipore; RRID:AB_177621) or anti-Mef2c (1:250; Abcam ab197070; RRID:AB_2629454) antibody followed by AlexaFluor-555 or AlexaFluor-488 conjugated secondary antibodies and dehydration. Cover slips were mounted using DPX mountant (Sigma). For Nissl staining, slices were

mounted on Superfrost Plus microscope slides (Fisher), dried, and dehydrated. Slices were stained for 6 min in 37°C cresyl violet (0.1%; Fisher), differentiated for 2 min, dehydrated (100% ethanol), and cleared (xylenes). Cover slips were mounted using DPX mountant (Sigma). Cortical thickness was assessed using ImageJ software (NIH).

## Immunoblotting of in vivo samples

Adult (8–12 week old) male mice were euthanized by $CO_2$ asphyxiation followed by decapitation. Tissue from different brain regions was rapidly dissected and frozen on dry ice. Tissues were sonicated on ice in a SDS lysis buffer: 1% (w/v) SDS, 300 mM sucrose, 10 mM NaF (Sigma), 50 mM HEPES (Sigma), and 1X Complete Protease Inhibitor cocktail (Roche). Samples were boiled for 10 min, then centrifuges at 16,000 x g for 10 min. Total protein concentration was determined by the DC protein assay kit (BioRad), and 20 µg of total protein was resolved using 10% SDS-PAGE. Proteins were transferred to Immobilon-FL PVDF (Millipore), blocked in Odyssey blocking buffer (Li-Cor) for 2 hr, and incubated overnight with either anti-Mef2c (AbCam ab197070; 1:2500; RRID:AB_2629454) or anti-Neuronal class III $\beta$-tubulin (Tuj1, Covance; 1:10,000; RRID:AB_2313773) antibodies. Blots were developed with enhanced chemiluminiscence western blotting detection reagent (Amersham ECL-Prime; GE Healthcare) or Odyssey CLx Western blot system (LiCor Biosciences).

## Dissociated cortical cultures

Primary cortical neurons were cultured from individual P0 mouse pups as previously described (*Beaudoin et al., 2012*), with modifications. Briefly, the neocortex was isolated from P0 individual pups (after removing hippocampus and midbrain), dissociated with 0.25% trypsin for 20 min, and plated on PDL (Sigma)- and laminin (Invitrogen)-coated 12 mm glass coverslips (Bellco) in a 24-well plate at 150,000 cells/well in Neurobasal media (NB) (Invitrogen) supplemented with 10% fetal bovine serum (FBS) (Invitrogen), 1% penicillin/streptomycin (P/S) (Sigma), and 1% L-glutamine (Q) (Sigma) and incubated at 37°C/5% $CO_2$ in a humidified incubator. Three to four hours after plating, the media was changed to NB supplemented with 2% B27 (Sigma), 1% P/S, and 1% Q. Every 3–4 days, half of the media was removed and replaced with fresh NB+B27+P/S+Q media.

## Transfection of primary cortical cultures

Dissociated cortical neurons were transfected at 5 days in culture (DIV) using calcium phosphate as previously described (*Flavell et al., 2006*). For synaptic staining experiments, neurons were transfected with myristolated-GFP and incubated till DIV 17–18.

## Luciferase assays in primary neurons

Dissociated cortical neurons were transfected at 5 DIV using calcium phosphate as previously described (*Flavell et al., 2006*). Cultures were stimulated for 5 hr using 60 mM KCl 42–48 hrs post-transfection and harvested for dual luciferase activity (Promega). For the luciferase assay, MEF2-response element (MRE)-firefly luciferase activity was divided by TK-renilla luciferase activity to control for transfection efficiency in each independent well.

To test effects of MEF2C-EN and MEF2C-VP16 on MEF2-dependent transcription, P0 cortical cultures were co-transfected using calcium phosphate method at DIV4 with a myristolated GFP, 3XMRE-mCherry, and either MEF2C-EN, MEF2C-VP16, or an empty vector control. After 72 hr, cultures were fixed with 4% PFA/Sucrose and imaged for mCherry expression. To quantify normalized MEF2 activity in each transfected neuron, 3XMRE-mCherry fluorescence intensity was divided by GFP signal intensity.

## Quantification of imaging and analysis

Image acquisition and quantification were performed in a blinded manner. Sixteen-bit images of neurons were acquired on a Leica SP8 confocal microscope using a 63x objective. Within each experiment, images were acquired with identical settings for laser power, detector gain, and amplifier offset. Images were acquired as a z-stack (5–20 optical sections and 0.5 µm step size). Maximum intensity projections were created from each stack.

For GAD65/GABA$_A$R γ2 experiments, synapse density was quantified as the overlap of GFP, α-GAD65 (Millipore MAB351, 1:1000; RRID:AB_11214081) and α-GABA$_A$R γ2 (Millipore AB5559,

1:100; RRID:AB_177523) staining using a custom ImageJ macro (see *Source code 1*, *2* and *3*). For each neuron, the threshold for GFP, GAD65 and GABARγ2 was determined from the sum of the average pixel intensity and standard deviation for each independent neuron. This thresholding method was then consistently applied across all images within the experiment. A binary mask including all pixels above the threshold was created for all channels for each image and the 'Analyze particles' function was used to determine regions of triple co-localization at least one pixel in size. To calculate synapse density, this number was divided by the area of the neuron as measured using the GFP mask minus the cell body (for dendritic synapse density) or using the GFP mask of only the cell body (for soma synapse density). Approximately 5–20 images from 2–3 separate coverslips were acquired and analyzed for each condition within an experiment for a total of at least two experiments. Synapse density values within each experiment were normalized to account for the variation in antibody staining and neuronal density from experiment to experiment. Within an experiment, the average synapse density value was obtained for the control and for experimental conditions. The normalized value of each experiment is the average experimental value divided by the average control value. For experiments involving quantification of inhibitory synapse density onto interneurons specifically, the observation of GAD65 expression in the cell body was used to identify and distinguish these interneurons from other excitatory cell types. Statistical significance was determined by t-test by using GraphPad Prism (RRID:SCR_002798). Error bars denote standard error.

## Dendritic spine analysis

Spine density was quantified as the number of spines divided by the length of a 15–20 µm stretch of dendrite. Spines from at least one secondary and one tertiary dendrite per image were manually counted using the 'Cell Counter' function in ImageJ software (NIH; RRID:SCR_003070). Statistical significance was determined by unpaired t-test by using GraphPad Prism (RRID:SCR_002798). Error bars denote standard error.

## Dendritic complexity

Images were acquired in a similar manner as described previously, except images were taken using a 20x objective. Maximum Intensity projections were generated from each stack and morphology was assessed using the 'Concentric Circles' plugin for ImageJ (NIH). The parameters for concentric circles plugin were set to generate 11 concentric circles at a line width of 1.0. X and Y values were set at the center of the soma of the transfected neuron and the inner radius and outer radii were calculated to produce a distance of 10 µm between circles. Dendritic morphology was then determined by manually counting the number of dendrite intersections per circle. The number of dendrite intersections per circle for each neuron within either the control group or experimental group were averaged together to generate an average number of intersections per radius for either control or experimental condition. Statistical difference was determined by using a two-way ANOVA with repeated measures in GraphPad Prism (RRID:SCR_002798). Error bars denote standard error.

## Brain slice preparation and electrophysiology

Acute neocortical slices of somatosensory, or 'barrel' cortex, were prepared from male or female MEF2C$^{fl/fl}$ or MEF2C$^{fl/fl}$; Cre$^{Emx1}$ littermates from age P20-25 (3 week) and bred on a mixed SVeV-129/C57BL/6J background. Mice were anesthetized with an I.P. injection of Ketamine (125 mg/kg)/Xylazine (25 mg/kg) and the brain removed. Coronal slices, 250–300 µm thick, were prepared in partially frozen dissection buffer consisting of (in mM): 110 choline chloride, 2.5 KCl, 1.25 $Na_2H_2PO_4$, 25 $NaHCO_3$, 25 D-glucose, 3.1 Na pyruvate, 11.6 Na ascorbate, 1 kynurenate, 7 $MgCl_2$, and 0.5 $CaCl_2$, aerated with 95% $O_2$ and 5% $CO_2$ prior to and during the slicing procedure. Slices for some experiments were prepared in 4°C dissection buffer consisting of (in mM): 75 sucrose, 87 NaCl, 3 KCl, 1.25 $NaH_2PO_4$, 7 $MgSO_4$, 26 $NaHCO_3$, 20 dextrose, and 0.5 $CaCl_2$, aerated with 95% $O_2$ and 5% $CO_2$. All solutions were pH 7.4. Genotypic differences using these different dissection solutions were the same so the results were pooled. For experiments in animals aged $\geq$P21, the mice were transcardially perfused with dissection buffer containing 1 mM kynurenic acid. Slices were then transferred to a 300 mOsM artificial cerebrospinal fluid (ACSF) solution containing in mM: 125 NaCl, 2.5 KCl, 1.25 $Na_2H_2PO_4$, 25 $NaHCO_3$, 10 D-glucose, 1 kynurenic acid, 2 $MgCl_2$, and 2 $CaCl_2$, to recover at 35°C for 25 min, and then transferred to room temperature (~21°C) for 30 min prior to recording.

Whole-cell recordings were performed in layer 2/3 neurons (resting $V_m < -50mV$, input resistance > 80 MΩ) centered above a barrel hollow, and cells were targeted with IR-DIC optics in an Olympus FV300 microscope. Recordings were performed at room temperature. Data were collected with a 10 kHz sampling rate and a 3 KHz Bessel filter.

## Miniature excitatory postsynaptic currents (mEPSCs)

Miniature excitatory postsynaptic currents were recorded in voltage clamp (at −70mV) in ACSF containing (in mM): 126 NaCl, 3 KCl, 1.25 $NaH_2PO_4$, 2 $MgSO_4$, 26 $NaHCO_3$, 25 dextrose, and 2 $CaCl_2$ with 1 μM tetrodotoxin (TTX), and 100 μM picrotoxin, to block mIPSCs. ACSF was aerated with 95% $O_2$ and 5% $CO_2$ and recycled. The internal solution contained in mM: 120 K-Gluconate, 5 NaCl, 10 HEPES, 1.1 EGTA, 4 MgATP, 0.4 $Na_2GTP$, 15 phosphocreatine, 2 $MgCl_2$, and 0.1 $CaCl_2$, pH 7.25 and 290 mOsm. The junction potential was ~10 mV and was not corrected.

## Miniature inhibitory postsynaptic currents (mIPSCs)

Miniature inhibitory postsynaptic currents were recorded in voltage clamp (at −70 mV) using a high-chloride internal solution containing in mM: 79 K-gluconate, 44 KCl, 6 NaCl, 10 HEPES, 0.2 EGTA, 4 MgATP, 0.4 $Na_2GTP$, 15 phosphocreatine, 2 $MgCl_2$, and 0.1 $CaCl_2$, which results in inward mIPSCs. To pharmacologically isolate mIPSCs, the extracellular ACSF contained 1 μM TTX, 5 μM CPP (NMDA-receptor antagonist), and 20 μM DNQX (AMPA-receptor antagonist).

## mPSC analysis

Miniature EPSCs and mIPSCs were analyzed using Mini Analysis (Synaptosoft) with the following parameters: amplitude threshold = 7 pA, area threshold = 10 pA. Events were automatically detected by the software, and non-events were then manually deleted upon visual inspection.

## UP state recordings

Persistent activity states or 'UP' states were measured in acute neocortical slices obtained from P18-24 mice as previously described (*Hays et al., 2011*). Briefly thalamocortical slices (400 μm) were perfused in ACSF containing (in mM) 126 NaCl, 3 KCl, 1.25 $NaH_2PO_4$, 26 NaHCO3, 2 $MgCl_2$, 2 $CaCl_2$, and 25 D-glucose for 1 hr at 32°C in an interface recording chamber, and then perfused for 45 min with the same ACSF supplemented with 5 mM KCl, 1 mM $MgCl_2$ and 1 mM $CaCl_2$, which mimics endogenous ionic concentrations. Spontaneously-generated UP-states were extracellularly recorded from layer 4 (L4) of the primary somatosensory cortex for 10 min using 0.5 MΩ tungsten microelectrodes, amplified 10,000-fold, sampled at 2.5 kHz, and filtered between 300 Hz and 5 kHz. All measurements were analyzed using custom Labview software. The beginning of an UP-state was defined as events in which the amplitude remained above threshold for at least 100 ms. The end of the UP-state was determined after the event amplitude decreased below threshold for >600 ms. Two events within 600 msec were defined as a single UP-state. All of the 20 slices prepared from MEF2C[fl/fl] mice displayed UP states, whereas only 9 of 18 slices from MEF2C[fl/fl]; Cre[Emx1] mice displayed UP states during a 10 min recording session, perhaps reflecting the reduced excitability of the MEF2C[fl/fl]; Cre[Emx1] circuits. Therefore, UP state duration and amplitude was only measured in the 9 slices that expressed UP states.

## Statistical analysis for electrophysiology

All data were analyzed using unpaired t-tests or 2-way ANOVA, as indicated, using GraphPad Prism (RRID:SCR_002798). *$p<0.05$, **$p<0.01$, ***$p<0.001$, ****$p<0.0001$.

## RNA Isolation and Reverse transcription PCR

Tissue from the somatosensory cortex of adult male mice were rapidly dissected and frozen at −80°C. Samples were thawed in TRIzol (Invitrogen), homogenized, and processed according to manufacturer's protocol. Total RNA was reverse-transcribed using Superscript III (Invitrogen) with random hexamers.

## RNA-Seq

Total RNA was isolated from somatosensory cortex as described above. Sequencing was performed by the Harvard Biopolymer Facility using PolyA mRNA isolation, directional RNA-seq library preparation, and the Illumina HiSeq2500 sequencer. Reads were aligned to mm9 using TopHat (*Trapnell et al., 2009*) (RRID:SCR_013035) and Bowtie (*Langmead et al., 2009*) (RRID:SCR_005476). Gene counts were calculated by HTSeq package (*Anders et al., 2015*) (RRID:SCR_005514) using the relative UCSC mm9 gtf file. Counts were normalized by RPKM (*Mortazavi et al., 2008*). We applied a treatment specific RPKM filtering considering genes with RPKM values more than 0.5 either in treatments or control. DESeq (*Anders and Huber, 2010*) (RRID:SCR_000154) was used to detect the differentially expressed genes (|logFC| > 0.3, FDR < 0.05). The RNA-Seq data discussed in this publication have been deposited in NCBI's Gene Expression Omnibus (RRID:SCR_005012) and are accessible through GEO Series accession number GSE87202 (http://www.ncbi.nlm.nih.gov/geo/query/acc.cgi?acc=GSE87202).

### Gene ontology

Gene ontology enrichment was performed using all of the expressed genes as background. We used DAVID (RRID:SCR_003033) with high stringency parameters (*Huang et al., 2009*), and we additionally confirmed the same GO categories with WebGestalt (RRID:SCR_006786) with similar approach (*Zhang et al., 2005*). DAVID adjusted p-values were used for further evaluation.

## RNA-Seq statistics

We assumed that the samples were normally distributed. *P*-values for overlaps were calculated with hypergeometric test using a custom made R script. We retained an independent background for population size (Allen brain expressed genes). P-values were subsequently adjusted for multiple comparisons using Benjamini-Hochberg FDR procedure. Two-way permutation test of 1000 was adapted to validate the overlaps. First we randomize the external gene sets (e.g ASD, ID, SynDB, FMRP) randomly selecting same number of genes from independent brain expressed genes list (MGI and/or Allen brain database) and subsequently calculating the overlap p-values. The second approach randomized the internal gene sets (e.g. DEGs gene set) randomly selecting same number of genes from RNA-seq expressed genes and subsequently calculating the overlap p-values. Moreover we adapted a permutation test to evaluate the detected DEG, randomizing 1000 times the RNA-seq data and recalculating the DEG. Analysis for RNA-seq were performed using custom made R scripts implementing functions and adapting statistical designs comprised in the libraries used.

## Mouse behavior testing

For all behavior tests, test mice were acclimated in a holding room outside the test room for 1 hr prior to testing. Behaviors in *Mef2c* cKO mice were compared to Cre-negative littermates tested on the same day. All behavioral tests were conducted using young adult male mice (8–12 weeks), except juvenile communication (USV) recordings. All behavior tests were conducted during the light-phase.

## Behavior data analysis

All data are presented as mean ± SEM. All comparisons were between littermates using appropriate two-sided statistical tests (specified in figure legends). We assumed that the samples were normally distributed. Outliers were determined using GraphPad's outlier calculator and excluded from analysis. *P*-values were calculated with unpaired *t*-test (two-tailed) or two-way ANOVA followed with Sidak's multiple comparisons post-hoc test using GraphPad Prism (RRID:SCR_002798), with specific test described in figure legends.

### Social interaction

For social vs novel object SI testing (*Figure 5—figure supplement 1K*), mice were acclimated for 5 min in an open field arena (44 cm$^2$) with 2 clear plexiglass holding chambers on either side of the arena. After acclimation, test mice were removed from arena, and a novel, conspecific mouse and a novel object (medium black paper binder) were placed in either holding chamber. Test mice were returned to arena and recorded for 5 min using Ethovision XT software (Noldus; RRID:SCR_000441).

Interaction time is defined as time spent in an 8 cm interaction zone around the holding chambers. For novel mouse SI (*Figure 5E*), the same paradigm and arena was used, except the novel object was not introduced. Preference index (*Figure 5E*) reflects (time interacting with social target – time interacting with empty holding cage) / (total time interacting with social target and empty cage).

## Social smell

Test mice were acclimated for 5 min in an open field arena with dry cotton-tipped applicators on opposite sides of the arena. Test mice were removed, and cotton-tipped applicators were dipped in either water or dirty bedding from a novel, conspecific mouse cage (social scent) and returned to the arena. Test mice were then returned to the arena and recorded for 5 min using Ethovision XT (Noldus; RRID:SCR_000441). Interaction time is defined as time spent in a 6 cm interaction zone around each cotton-tipped applicator.

## Nest building

Nest building was performed as previously described (*Deacon, 2006*). Briefly, test mice are singly housed, and a new nestlet is introduced to the cage 1 hr before the active phase (dark phase). The next morning, the remaining compacted cotton from the nestlet is weighed, and the nests are scored on a rating scale of 1–5 (*Deacon, 2006*), with experimenter blinded to genotype. A well-structured nest is scored 5 and failure to disturb the nestlet is scored 1.

## Tube test for social dominance

Mice were introduced into each end of a white PVC pipe (2.5 cm diameter; 30.5 cm length) and simultaneously released. The trial ended when one mouse completely backed out of the tube (during a 2 min trial). Matches were considered a draw if neither animal retreated after 2 min and were excluded from analysis. Mice were matched according to body weight and paired from different cages. Five consecutive trials were conducted with each pair.

## Ultrasonic vocalization recordings

Social ultrasonic vocalizations (USVs) were recorded from adult mice as previously described (*Ey et al., 2013*). Briefly, ovariectomized female mice (C57BL/6; Jackson Labs) were injected with 15-µg estradiol 48 hr prior to testing and 1-mg progesterone 4 hr before testing to induce estrous. Test mice (8–12 week old male mice) were acclimated to a clean home-cage in a sound attenuated chamber for 5 min. After acclimation, an estrous female is introduced into the holding chamber with the male test mouse, and USVs were recorded for 5 min using Avisoft UltraSoundGate equipment (UltraSoundGate 116Hb with Condenser Microphone CM16; Avisoft Bioacoustics, Germany). USVs were analyzed using Avisoft SASLab Pro (Avisoft Bioacoustics) using a 20 kHz cutoff. USVs were categorized as previously described (*Ey et al., 2013*) by a trained experimenter blinded to genotypes to prevent bias. <u>Distress USVs</u> were recorded from juvenile mice (pups) as previously described (*Ey et al., 2013*; *Scattoni et al., 2008*). Briefly, individual pups of both sexes were identified with long-lasting subcutaneous tattoos (green tattoo paste; Ketchum) on the paws on post-natal day 3 (P3). Pups of both sexes were recorded in a random order in a small, sound-attenuated chamber following separation from dam and littermates. USVs were recorded for 3 min on post-natal days 4, 6, and 10. USVs were analyzed and characterized as previously described for the adult USVs using Avisoft SASLab Pro (Avisoft Bioacoustics, German).

## Sucrose preference

Test mice were singly housed and provided 2 identical ball-bearing sipper-style bottles to drink from. Mice were acclimated to the 2 bottles for 4 days, where both bottles contained water on days 1 and 3 or sucrose (or quinine) solution on days 2 and 4. On days 5–8, mice were presented with 2 bottles, one with water and one with sucrose (1% (w/v) or 4% (w/v)) or quinine (0.04% (w/v)). Daily, the consumption of water and sucrose/quinine was measured, and the bottle position was altered to avoid potential side bias (*Renthal et al., 2007*). Data is presented as (solution consumption – water consumption) / total consumption = Preference Index.

## Repetitive behaviors

Test mice were introduced into an activity test chamber within a photobeam activity system (MedAssociates), where 2 rows of photobeams measure the mouse's horizontal and vertical activity. Activity was monitored for 1 hr using MedAssociates Activity Monitor (V.5) (RRID:SCR_014296), and data is presented as time spent jumping and time spent in fine motor movements (stereotypy), as defined by non-ambulatory motor movements (thoroughly described in Activity Monitor V.5 handbook). To assess grooming and digging, mice were placed in a clean home cage with bedding, and activity was recorded for 30 min using video cameras. Time spent grooming and digging were recorded by an experimenter that was blinded to the genotypes.

## Locomotor activity

Test mice were placed in a clean home cage with minimal bedding inside the Photobeam Activity System (San Diego Instruments), where 5 photobeams measure the mouse's locomotor activity in 5-minute bins. Activity was monitored over the course of an hour, and data is presented as number of beam breaks/5 min during the hour of recording and total number of beam breaks.

## Rotarod test

Mice are placed on the roller in a Rotarod apparatus (Stoelting; Ugo Basile Apparatus) for a 5-minute training session with a rotation of 4 rpm, and replaced if test mouse falls during training session. After the training session, test mice are returned to the roller, where the speed steadily increases from 5–40 rpm, and the latency for the mouse to fall off the roller is recorded. Each animal receives two testing sessions.

## Fear conditioning test

Fear conditioning was performed as previously described (*Wehner and Radcliffe, 2004*). Briefly, test mice are placed in a fear conditioning chamber (MedAssociates) and allowed to explore the arena for 2 min, after which a loud auditory stimulus (30 s; 90 dB) that co-terminates with a 2 s mild footshock (0.5 mA) is presented to the animal. This is repeated twice more, with a 1 min interval separating the tones/shocks. After 24 hr, animals are returned to the chamber, and its behavior in the context, in a new context, and with the audible tone played in a new context is recorded by a video-tracking system (Video Freeze V2.6; MedAssociates) (RRID:SCR_014574). Data are presented as percent of time the mouse is immobile.

## Shock response

Shock sensitivity was measured using MedAssociates Inc Startle Reflex System and Advanced Startle software program. Mice were placed in Plexiglas and wire grid animal holders (ENV-246C) attached to a load cell platform (PHM-250) contained within a sound-attenuated chamber. Footshocks (0.1 – 0.5 mA) were delivered by Stand Alone Stimulators/Scramblers (ENV-414) connected to the wire grid floors of the animal holder. Displacements of the load cell stabilimeter were converted into arbitrary units by an analog-to-digital converter (ANL-925C Amplifier) interfaced to a personal computer.

### Behavior data analysis

All data are presented as mean ± SEM. All comparisons were between littermates using appropriate two-sided statistical tests (specified in Figure legends). We assumed that the samples were normally distributed. Outliers were determined using GraphPad's outlier calculator and excluded from analysis. *P*-values were calculated with unpaired *t*-test (two-tailed) or two-way ANOVA followed with Sidak's multiple comparisons post-hoc test using GraphPad Prism, with specific test described in Figure legends.

## Acknowledgements

The authors thank the research community within the basic neuroscience division at McLean Hospital, Drs. Shari Birnbaum and Craig Powell for providing equipment and technical advice, the Harvard NeuroDiscovery Center for performing rotarod test, and Dr. Eric Olson (UTSW) for sharing the

floxed *Mef2c* mice. AH was supported by a NIH F32 award (F32 HD078050) and a Brooking Fellowship (McLean Hospital). This work was also supported in part by grants from the Simons Foundation (SFARI #206919 to CC and KH), the Ellison Foundation of Boston (to CC), and the Lurie Center for Autism (to CC), the NIH (DA027664 to CC and HD052731 to KH), and a NIH small instrument grant (OD010737).

## Additional information

### Funding

| Funder | Grant reference number | Author |
|---|---|---|
| National Institutes of Health | F32 HD078050 | Adam J Harrington |
| Simons Foundation | SFARI #206919 | Kimberly M Huber<br>Christopher W Cowan |
| National Institutes of Health | HD052731 | Kimberly M Huber |
| National Institutes of Health | OD010737 | Kimberly M Huber |
| National Institutes of Health | DA027664 | Christopher W Cowan |

The funders had no role in study design, data collection and interpretation, or the decision to submit the work for publication.

### Author contributions
AJH, AR, JK, Conception and design, Acquisition of data, Analysis and interpretation of data, Drafting or revising the article; KR, KL, Acquisition of data, Analysis and interpretation of data, Drafting or revising the article; SB, CWC, Conception and design, Analysis and interpretation of data, Drafting or revising the article; GM, GK, KMH, Analysis and interpretation of data, Drafting or revising the article; JR, YG, Acquisition of data, Drafting or revising the article

### Author ORCIDs
Christopher W Cowan, http://orcid.org/0000-0001-5472-3296

### Ethics
Animal experimentation: This study was performed in strict accordance with the recommendations in the Guide for the Care and Use of Laboratory Animals of the NIH. All of the animals were handled according to approved institutional animal care and use committee (IACUC) protocols (#2015N000178 and #2015N000160) of McLean Hospital.

## Additional files

### Supplementary files
• Source code 1. ImageJ macro for quantification of colocalized pre- and post-synaptic marker puncta on a GFP-filled neuron mask.

• Source code 2. ImageJ macro for quantification of presynaptic marker puncta on a GFP-filled neuron mask.

• Source code 3. ImageJ macro for quantification of postsynaptic marker puncta on a GFP-filled neuron mask.

• Supplementary file 1. Gene expression in SSC of *Mef2c* cKO mice.

### Major datasets
The following dataset was generated:

| Author(s) | Year | Dataset title | Dataset URL | Database, license, and accessibility information |
|---|---|---|---|---|
| Harrington AJ, Raissi A, Rajkovich K, Berto S, Kumar J, Molinaro G, Raduazzo J, Guo Y, Loerwald K, Konopka G, Huber K, Cowan CW | 2016 | MEF2C regulates cortical inhibitory and excitatory synapses and behaviors relevant to neurodevelopmental disorders | http://www.ncbi.nlm.nih.gov/geo/query/acc.cgi?acc=GSE87202 | Publicly available at the NCBI Gene Expression Omnibus (accession no: GSE87202). |

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
