## [Decision Letter]

Thank you for submitting your article "MEF2C regulates cortical inhibitory and excitatory synapses and behaviors relevant to neurodevelopmental disorders" for consideration by *eLife*. Your article has been favorably evaluated by a Senior Editor and two reviewers, one of whom is a member of our Board of Reviewing Editors. The reviewers have opted to remain anonymous.

The reviewers have discussed the reviews with one another and the Reviewing Editor has drafted this decision to help you prepare a revised submission.

Summary:

In this manuscript by Harrington et al., the authors analyzing the impact of loss of *Mef2c* on synaptic connectivity, cortical function and behavior. Using a large set of genetic, molecular, electrophysiological and behavioral experiments the authors convincingly demonstrate a quite specific function of this transcription factor in regulating neuronal network function and behavior. They found much less cortical spontaneous UP states, small decreases in excitatory drive and large increases in inhibitory drive. Those effects were in a cell autonomous fashion.

Interestingly, glutamatergic pyramidal cells (that lack MEF2C) show up to twofold increases in GABA synapse numbers, clearly arguing for a postsynaptic effect, Furthermore the effect was specific for glutamatergic neurons. MEF2C can act as repressor or activator, and the authors analysed using MEF2C mutants that its role in suppression is most relevant for the morphological phenotype seen.

Subsequent RNA seq analysis to identify potential targets of MEF2C that may mediate synapse regulation show an overrepresentation of ASD and synapse linked genes being dysregulated, and the list of genes with altered expression provide some further hints which genes may be responsible for the differential role of Glu/GABA synapse regulation, which should yield in a set of interesting follow-up experiments.

Finally, the authors perform an extensive set of behavioral experiments that round this work up to demonstrate the putative mechanistic participation of MEF2C in ASD disorders.

Overall the work is well done, and the manuscript is well written. The extension of MEF2 family functions to inhibitory synapses is novel as is the observation that it mediates these synaptic developmental effects via its repressor functions. Also the findings that MEF2C and in conclusion also the other isoforms of MEF2 have unique phenotypes is clearly interesting for the field of development and associated disorders. The effect on GABA synapse number regulation in a cell type specific manner highlights new biology provided by this study.

Essential revisions:

Related to the experiments of Figure 3, were these experiments done in culture? Please state how the input neuron was determined to be a pyramidal cell or a GABAergic interneuron, an unambiguous definition of cell type is quite critical for the conclusions.

---

## [Author Response]

*Essential revisions:*

*Related to the experiments of Figure 3, were these experiments done in culture? Please state how the input neuron was determined to be a pyramidal cell or a GABAergic interneuron, an unambiguous definition of cell type is quite critical for the conclusions.*

Yes, the experiments in Figure 3 were done in cultured primary cortical neurons generated from *Mef2c* WT or floxed mutant mice. The input neuron was characterized based on cellular morphology and GAD-65 immunostaining patterns (GABAergic interneurons show high GAD-65 staining on neuronal cell bodies (somas)). We have now included descriptions on how these neurons were determined in the manuscript (Results, subsection “MEF2C regulates both excitatory and inhibitory synapses in a cell-autonomous manner”, second paragraph; Methods section).

Additional manuscript edit request: In our original submission, we excluded supporting data demonstrating that the MEF2C-EN and MEF2C-VP16 repressed or enhanced, respectively, MEF2-dependent transcription in the cultured cortical neurons (although this basic finding has been published by many others in various cell types). While not requested by reviewers, we request that this data and sub-figure be included in the final published manuscript. We have added the new data plots in Figure 3—figure supplement 3F, and now refer to it in the Results section (subsection “MEF2C regulates both excitatory and inhibitory synapses in a cell-autonomous manner”, last paragraph) and in the Methods section). We feel that this data strengthens the findings by confirming the functions of these MEF2 fusion proteins under our experimental conditions.

In addition, we have now uploaded our RNA-Seq data files from control and *Mef2c cKO* mice into the NCBI public functional genomics data repository GEO (NCBI Gene Expression Omnibus). We have added the reference to the Materials and Methods section “RNA-Seq”.